# Self-Improving Robots: End-to-End Autonomous Visuomotor Reinforcement Learning

**Archit Sharma**[*]    **Ahmed M. Ahmed**[*]    **Rehaan Ahmad**    **Chelsea Finn**
Stanford University

**Abstract:** In imitation and reinforcement learning (RL), the cost of human supervision limits the amount of data that the robots can be trained on. While RL offers a framework for building self-improving robots that can learn via trial-and-error autonomously, practical realizations end up requiring extensive human supervision for *reward function design* and *repeated resetting* of the environment between episodes of interactions. In this work, we propose MEDAL++, a novel design for self-improving robotic systems: given a small set of expert demonstrations at the start, the robot autonomously practices the task by learning to both *do* and *undo* the task, simultaneously inferring the reward function from the demonstrations. The policy and reward function are learned end-to-end from high-dimensional visual inputs, bypassing the need for explicit state estimation or task-specific pre-training for visual encoders used in prior work. We first evaluate our proposed system on a simulated non-episodic benchmark, EARL, finding that MEDAL++ is both more data efficient and gets up to 30% better final performance compared to state-of-the-art vision-based methods. Our real-robot experiments show that MEDAL++ can be applied to manipulation problems in larger environments than those considered in prior work, and autonomous self-improvement can improve the success rate by 30% to 70% over behavioral cloning on just the expert data. Code, training and evaluation videos along with a brief overview is available at: https://architsharma97.github.io/self-improving-robots/

**Keywords:** reinforcement learning, autonomous, reset-free, manipulation

## 1    Introduction

While imitation learning methods have shown promising evidence for generalization via large-scale teleoperated data collection efforts [1, 2], human supervision is expensive and collected datasets are still incommensurate for learning robust and broadly performant control. In this context, the aspirational notion of *self-improving* robots becomes relevant: robots that can learn and improve from their own interactions with the environment autonomously. Reinforcement learning (RL) is a natural framework for such self-improvement, where the robots can learn from trial-and-error autonomously in principle. However, RL deployment requires domain expertise and extensive supervision in practice for state estimation, designing reward functions, and repeated resetting of the environments after every episode of interaction.

In particular, the human supervision for repeatedly resetting the environment through training is an impediment to

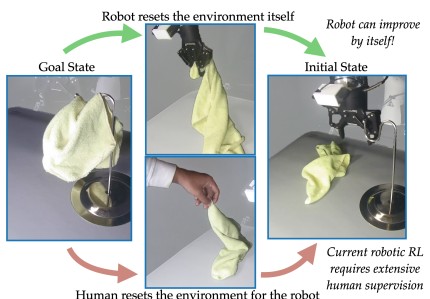

Figure 1: A robot resets the environment from the goal state to the initial state (*top*), in contrast to a human resetting the environment for the robot (*bottom*). While latter is the norm in robotic reinforcement learning, a robot that can reset the environment and practice the task autonomously can train on more data, and thus, be more competent.

---

[*]Authors contributed substantially to real-robot results. Correspondence to architsh@stanford.edu.

7th Conference on Robot Learning (CoRL 2023), Atlanta, USA.

building autonomous robots, visualized in Figure 1. Several prior works [3, 4, 5] show that standard RL algorithms can fail catastrophically when the reset frequency is decreased. While recent works address the lack of supervision for repeated resetting [4, 6, 7, 8, 9], learning efficiently in the absence of frequent environment resets remains a challenge in the real-world. Task-relevant states constitute only a small subset of all possible states, especially in larger real-world environments, and the robot can drift far from these task-relevant states when learning without repeated resets. In view of this, using a small set of demonstrations can be an effective choice to construct self-improving systems. Expert demonstrations can alleviate challenges related to exploration [10] and enable efficient autonomous RL by encouraging agent to stay close to task-relevant states in the demonstrations [11]. And since the human supervision required for collecting the demonstrations is *front-loaded*, i.e., before the training begins, the robot can collect data autonomously and self-improve thereon. Importantly, we make the observation that the terminal states in expert trajectories effectively represent the goal distribution. This allows us to learn a goal-reaching reward function without any additional data collection or reward engineering.

In this work, our main contribution is MEDAL++, a carefully designed system that can train autonomously in the real-world with minimal task-specific engineering. MEDAL++ builds upon [11] with several crucial components that enable efficient end-to-end autonomous training in the real-world: First, we learn an encoder for high-dimensional visual inputs end-to-end using DrQ-v2 [12], bypassing the need for state estimation or task-specific pre-training of visual encoders. Second, we judiciously use expert demonstrations by learning a goal-reaching reward function [13, 14], eliminating the need for engineered reward functions. Finally, we improve the sample efficiency by using an ensemble of $Q$-value functions and increasing the update steps per sample collected [15], using BC regularization on expert data to regularize policy learning towards demonstration data [16], and oversampling transitions from demonstration data when training the $Q$-value function [10]. We evaluate MEDAL++ on a pixel-based control version of EARL [5], a non-episodic learning benchmark and observe that MEDAL++ is more data efficient and gets up to 30% better performance compared to competitive methods [4, 11]. Most importantly, we conduct real-robot evaluations using a Franka Panda robot arm on four manipulation tasks, such as hanging a cloth on a hook, covering a bowl with a cloth and peg insertion, all from RGB image observations. We observe that MEDAL++ can improve the success rate of the policy by 30% to 70% over a behavioral cloning policy learned only on the expert data, suggesting that MEDAL++ is a step towards self-improving robotic systems.

## 2 Related Work

Several works have demonstrated the emergence of complex skills on a variety of problems using reinforcement learning on real robots [17, 18, 19, 20, 21, 22, 23, 24, 25, 26, 27], However, these prior works require the environment to be reset to a (narrow) set of initial states for every episode of interaction with the environment. Such resetting of the environment either requires repeated human interventions and constant monitoring [28, 29, 30, 31, 32] or scripting behaviors [19, 23, 33, 34, 24, 35] which can be time-intensive while resulting in brittle behaviors. Some prior works have also designed the task and environment to bypass the need for resetting the environment [36, 37, 21], but this applies to a restricted set of tasks.

More recent works have identified the need for algorithms that can work autonomously with minimal supervision required for resetting the environments [38, 4, 5]. Several recent works propose to learn a backward policy to undo the task, in addition to learning a forward policy that does the task [38, 6, 7, 8, 39, 40]. In this work, we build upon MEDAL [11], where the backward policy learns to match the distribution of states visited by an expert to solve the task. While the results from these prior papers are restricted to simulated settings, some recent papers have demonstrated autonomous training on real robots [4, 41, 42, 43]. However, the results on real robots have either relied on state estimation [41, 42], pre-specified reward functions [43] or task-specific decomposition into subgoals [44]. R3L [4] also considers the setting of learning from image observations without repeated resets and specified reward functions, similar to our work. It uses a backward policy that optimizes for state-novelty while learning the reward function from a set of goal images collected prior to training [14]. However, R3L relies on frozen visual encoders trained independently on data

collected in the same environment, and optimizing for state-novelty does not scale to larger environments, restricting their robot evaluations to smaller, easier to explore environments. Our simulation results indicate that MEDAL++ learns more efficiently than R3L, and real robot evaluations indicate the MEDAL++ can be used on larger environments. Overall, our work proposes a system that can learn end-to-end from visual inputs without repeated environment resets, with real-robot evaluations on four manipulation tasks.

## 3 Preliminaries

**Problem Setting**. We consider the autonomous RL problem setting [5]. We assume that the agent is in a Markov Decision Process represented by $(\mathcal{S}, \mathcal{A}, \mathcal{T}, r, \rho_0, \gamma)$, where $\mathcal{S}$ is the state space, potentially corresponding to high-dimensional observations such as RGB images, $\mathcal{A}$ denotes the robot's action space, $\mathcal{T} : \mathcal{S} \times \mathcal{A} \times \mathcal{S} \to \mathbb{R}_{\geq 0}$ denotes the transition dynamics of the environment, $r : \mathcal{S} \times \mathcal{A} \to \mathbb{R}$ is the (unknown) reward function, $\rho_0$ denotes the initial state distribution, and $\gamma$ denotes the discount factor. The objective is to learn a policy that maximizes $\mathbb{E}\left[\sum_{t=0}^{\infty} \gamma^t r(s_t, a_t)\right]$ when *deployed* from $\rho_0$ during *evaluation*. There are two key differences from the standard episodic RL setting: First, the training environment is non-episodic, i.e., the environment does **not** periodically reset to the initial state distribution after a fixed number of steps. Second, the reward function is not available during training. Instead, we assume access to a set of demonstrations collected by an expert prior to robot training. Specifically, the expert collects a small set of forward trajectories $\mathcal{D}_f^* = \{(s_i, a_i) \dots\}$ demonstrating the task and similarly, a set of backward demonstrations $\mathcal{D}_b^*$ undoing the task back to the initial state distribution $\rho_0$.

**Autonomous Reinforcement Learning via MEDAL**. To enable a robot to practice the task autonomously, MEDAL [11] trains a forward policy $\pi_f$ to solve the task, and a backward policy $\pi_b$ to undo the task. The forward policy $\pi_f$ executes for a fixed number of steps before the control is switched over to the backward policy $\pi_b$ for a fixed number of steps. Chaining the forward and backward policy reduces the number of interventions required to reset the environment. The forward policy is trained to maximize $\mathbb{E}\left[\sum_{t=0}^{\infty} \gamma^t r(s_t, a_t)\right]$, which can be done via any RL algorithm. The backward policy $\pi_b$ is trained to minimize the Jensen-Shannon divergence $\mathrm{JS}(\rho^b(s) \;||\; \rho^*(s))$ between the stationary state-distribution of the backward policy $\rho^b$ and the state-distribution of the expert policy $\rho^*$. By training a classifier $C_b : \mathcal{S} \mapsto (0, 1)$ to discriminate between states visited by the expert (i.e. $s \sim \rho^*$) and states visited by $\pi_b$ (i.e., $s \sim \rho^b$), the divergence minimization problem can be rewritten as $\max_{\pi_b} -\mathbb{E}[\sum_{t=0}^{\infty} \gamma^t \log\left(1 - C_b(s_t)\right)]$ [11]. The classifier used in the reward function for $\pi_b$ is trained using the cross-entropy loss, where the states $s \in \mathcal{D}_f^*$ are labeled 1 and states visited by $\pi_b$ online are labeled 0, leading to a minimax optimization between $\pi_b$ and $C_b$.

**Learning Reward Functions with VICE**. Engineering rewards can be tedious, especially when only image observations are available. Since the transitions from the training environment are not labeled with rewards, the robot needs to learn a reward function for the forward policy $\pi_f$. In this work, we consider VICE [45], particularly, the simplified version presented by Singh et al. [14] that is compatible with off-policy RL. VICE requires a small set of states representing the desired outcome (i.e., goal images) prior to training. Given a set of goal states $\mathcal{G}$, VICE trains a classifier $C_f : \mathcal{S} \mapsto (0, 1)$ where $C_f$ is trained using the cross entropy loss on states $\in \mathcal{G}$ labeled as 1, and states visited by $\pi_f$ labeled as 0. The policy $\pi_f$ is trained with a reward function of $\log C_f(s) - \log\left(1 - C_f(s)\right)$, which can be viewed as minimizing the KL-divergence between the stationary state distribution of $\pi_f$ and the goal distribution [46, 47, 48]. VICE has two benefits over pre-trained frozen classifier-based rewards: first, the negative states do not need to be collected by a person and second, the VICE classifier is harder to exploit as the online states are iteratively added to the label 0 set, continually improving the goal-reaching reward function implicitly.

## 4 MEDAL++: Practical and Efficient Autonomous Reinforcement Learning

The goal of this section is to develop a RL method that can learn from autonomous online interaction in the real world, given just a (small) set of forward $\mathcal{D}_f^*$ and backward demonstrations $\mathcal{D}_b^*$ without reward labels. Particularly, we focus on design choices that make MEDAL++ viable in the real

world in contrast to MEDAL: First, we describe how to learn from visual inputs without explicit state estimation. Second, we describe how to learn a reward function from the expert demonstrations to eliminate the need for ground truth rewards when training the forward policy $\pi_f$. Third, we describe the algorithmic modifications for training the $Q$-value function and the policy $\pi$ more efficiently. Finally, we describe how to construct MEDAL++ using all the components described here, training a forward policy $\pi_f$ and a backward policy $\pi_b$ to learn autonomously.

**Encoding Visual Inputs**. We embed the high-dimensional RGB images into a low-dimensional feature space using a convolutional encoder $\mathcal{E}$. The RGB images are augmented using random crops and shifts (up to 4 pixels) to regularize $Q$-value learning [12]. While some prior works incorporate explicit representation learning losses for visual encoders [49, 50], Yarats et al. [12] suggest that regularizing $Q$-value learning using random crop and shift augmentations is both simpler and efficient, allowing end-to-end learning without any explicit representation learning objectives. Specifically, the training loss for $Q$-value function on an environment transition $(s, a, s', r)$ can be written as:

$$\ell(Q, \mathcal{E}) = \Big(Q\left(\mathcal{E}\left(\text{aug}(s)\right), a\right) - r - \gamma \bar{V}\big(\mathcal{E}(\text{aug}(s'))\big)\Big)^2 \tag{1}$$

where $\text{aug}(\cdot)$ denotes the augmented image, and $r + \gamma \bar{V}(\cdot)$ is the TD-target. Equation 2 describes the exact computation of $\bar{V}$ using slow-moving target networks $\bar{Q}$ and the current policy $\pi$.

**Learning the Reward Function**. To train a VICE classifier $C_f$, we need to specify a set of goal states that can be used as positive samples. Instead of collecting a separate set of goals, we observe that the last $K$ states of every trajectory in $\mathcal{D}_f^*$ approximate the goal distribution, and thus, can be used as the goal set $\mathcal{G}$. The trajectories collected by the robot's policy $\pi_f$ will be used to generate negative states for training $C_f$. The policy is trained to maximize $-\log\left(1 - C_f(\cdot)\right)$ as the reward function, encouraging the policy to reach states that have a high probability of being labeled 1 under $C_f$, and thus, similar to the states in the set $\mathcal{G}$. The reward signal from the classifier can be sparse if the classifier has high accuracy on distinguishing between the goal states and states visited by the policy. Since the classification problem for $C_f$ is easier than the goal-matching problem for $\pi_f$, especially early in the training when the policy is not as successful, it becomes critical to regularize the discriminator $C_f$ [51]. We use spectral normalization [52], mixup [53] to regularize $C_f$, and apply random crop and shift augmentations to input images to create a broader training distribution.

As we have access to expert demonstrations $\mathcal{D}_f^*$, why do we match the policy's *state* distribution to the goal distribution, instead of GAIL [13, 54], which matches policy's *state-action* distribution to that of the expert? In a practical robotic setup, actions demonstrated by an expert during tele-operation and optimal actions for a learned neural network policy will be different. The forward pass through a policy network introduces a delay, especially as the visual encoder $\mathcal{E}$ becomes larger. Matching both the state and actions to that of the expert, as is the case with GAIL, can lead to sub-optimal policies and be infeasible in general. In contrast, VICE allows the robotic policies to choose actions that are different from the expert as long as they lead to a set of states similar to those in $\mathcal{G}$. The exploratory benefits of matching the actions can be recovered, as described in the next section.

**Improving the Learning Efficiency**. To improve the learning efficiency over MEDAL, we incorporate several changes in how we train the $Q$-value function and the policy $\pi$. First, we train an ensemble of $Q$-value networks $\{Q_n\}_{n=1}^N$ and corresponding set of target networks $\{\bar{Q}_n\}_{n=1}^N$. When training an ensemble member $Q_n$, the target is computed by sampling a subset of target networks, and taking the minimum over the subset. The target value $\bar{V}(s')$ in Eq 1 can be computed as

$$\bar{V}(s') = \mathbb{E}_{a' \sim \pi} \min_{j \in \mathcal{M}} \bar{Q}_j(s', a'), \tag{2}$$

where $\mathcal{M}$ is a random subset of the index set $\{1 \ldots N\}$ of size $M$. Randomizing the subset of the ensemble when computing the target allows more gradient steps to be taken to update $Q_n$ on $\ell(Q_n, \mathcal{E})$ [15] without overfitting to a specific target value, increasing the overall sample efficiency of learning. The target networks $\bar{Q}_n$ are updated as an exponential moving average of $Q_n$ in the weight space over the course of training. At iteration $t$, $\bar{Q}_n^{(t)} \leftarrow \tau Q_n^{(t)} + (1 - \tau)\bar{Q}_n^{(t-1)}$, where $\tau \in (0, 1]$ determines how closely $\bar{Q}_n$ tracks $Q_n$.

Importantly, we want to leverage the expert demonstrations to counteract the exploration challenge, especially because the training signal from VICE reward can be sparse. $Q$-value networks are typically updated on minibatches sampled uniformly from a replay buffer $\mathcal{D}$. However, the transitions in the demonstrations are generated by an expert, and thus, can be more informative about the actions for reaching successful states [10]. To bias the data towards the expert distribution, we oversample transitions from the expert data such that for a batch of size $B$, $\rho B$ transitions are sampled from the expert data uniformly and $(1 - \rho)B$ transitions are sampled from the replay buffer uniformly for $\rho \in [0, 1)$. Finally, we regularize the policy learning towards expert actions by introducing a behavior cloning loss in addition to maximizing the $Q$-values [16, 10]:

$$\mathcal{L}(\pi) = \mathbb{E}_{s \sim \mathcal{D}, a \sim \pi(\cdot | s)} \left[ \frac{1}{N} \sum_{n=1}^{N} Q_n(\mathcal{E}(\mathrm{aug}(s)), a) \right] + \lambda \mathbb{E}_{(s^*, a^*) \sim \rho^*} \left[ \log \pi \left( a^* \mid \mathcal{E}(\mathrm{aug}(s^*)) \right) \right],$$

where $\lambda \geq 0$ denotes the hyperparameter controlling the effect of BC regularization. Note that the parameters of the encoder are frozen with respect to $\mathcal{L}(\pi)$, and are only trained through $\ell(Q_n, \mathcal{E})$.

**Putting it Together: MEDAL++.** MEDAL++ trains a forward policy that learns to solve the task and a backward policy that learns to undo the task towards the expert state distribution. The parameters and data buffers for the forward policy are represented by the tuple $\mathcal{F} \equiv \left( \pi_f, \mathcal{E}^f, \{Q_n^f\}_{n=1}^N, \{\bar{Q}_n^f\}_{n=1}^N, C_f, \mathcal{D}_f^*, \mathcal{D}_f, \mathcal{G}_f \right)$, where the symbols retain their meaning from the previous sections. Similarly, the parameters and data buffers for the backward policy are represented by the tuple $\mathcal{B} \equiv \left( \pi_b, \mathcal{E}^b, \{Q_n^b\}_{n=1}^N, \{\bar{Q}_n^b\}_{n=1}^N, C_b, \mathcal{D}_b^*, \mathcal{D}_b, \mathcal{G}_b \right)$. Noticeably, $\mathcal{F}$ and $\mathcal{B}$ have a similar structure: Both $\pi_f$ and $\pi_b$ are trained using with $-\log(1 - C(\cdot))$ as the reward function (using their respective classifiers $C_f$ and $C_b$), with both classifiers trained to discriminate between the states visited by the policy and their target states. The primary difference is the set of positive target states $\mathcal{G}_f$ and $\mathcal{G}_b$ used to train $C_f$ and $C_b$ respectively, visualized in Figure 8. The VICE classifier $C_f$ is trained to predict the last $K$ states of every trajectory from $\mathcal{D}_f^*$ as positive, whereas we train the MEDAL classifier $C_b$ to predict all the states of forward demonstrations **except** the last $K$ states as positive. Optionally, we can also include the last $K$ states of backward demonstrations from $\mathcal{D}_b^*$ as positives for training $C_b$. The pseudocode for training is given in Algorithm 1, and more detailed description is available in Appendix A.1. Some key details: the control switches between the forward $\pi_f$ and the backward policy $\pi_b$ after a fixed number of steps. When executing in the real world, humans are allowed to intervene and reset the environment intermittently, switching the control over to $\pi_f$ after the intervention to restart the forward-backward cycle.

## 5 Experiments

The goal of our experiments is to determine whether MEDAL++ can be a practical method for self-improving robotic systems. We benchmark MEDAL++ against competitive methods [11, 4] on EARL [5] benchmark for non-episodic RL to evaluate the learning efficiency from high-dimensional observations, in Section 5.1. Our primary experiments in Section 5.2 evaluate MEDAL++ on four real robot manipulation tasks, primarily tasks with soft-body objects such as hanging a cloth on a hook and covering a bowl with cloth. The real robot evaluation considers the question of whether self-improvement is feasible via MEDAL++, and if so, how much self-improvement can MEDAL++ obtain? Finally, we run ablations to evaluate the contributions of different components to MEDAL++ in Section 5.3.

### 5.1 Benchmarking MEDAL++ on EARL

First, we benchmark MEDAL++ on continuous-control environment from EARL against state-of-the-art non-episodic autonomous RL methods. To be consistent with the benchmark, we use the ground truth reward functions for all the environments.

**Environments**. We consider three sparse-reward continuous-control environments from EARL benchmark [5], shown in Appendix, Fig 6. *Tabletop organization* is a simplified manipulation environment where a gripper is tasked to move the mug to one of the four coasters from a wide set of initial states, *sawyer door closing* task requires a sawyer robot arm to learn how to close a door starting from various positions, and finally the *sawyer peg insertion* task requires the sawyer robot

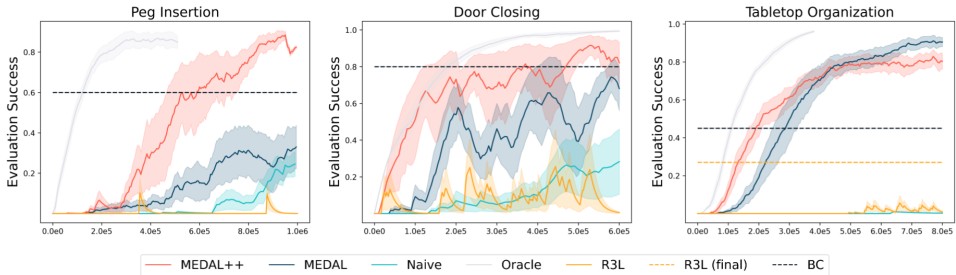

Figure 2: Comparison of autonomous RL methods on vision-based manipulation tasks in simulated environments from EARL [5]. MEDAL++ is both more efficient and learns a similarly or more successful policy compared to other methods.

arm to grasp the peg and insert it into a goal. Not only does the robot have to learn how to do the task (i.e. close the door or insert the peg), but it has to learn how to undo the task (i.e. open the door or remove the peg) to try the task repeatedly in a non-episodic training environment. All tasks are setup to return $84 \times 84$ RGB observations with sparse goal reaching reward functions. The training environment is reset to $s_0 \sim \rho_0$ every 25,000 steps of interaction with the environment. This is extremely infrequent compared to episodic settings where the environment is reset to the initial state distribution every 200-1000 steps. EARL comes with 5-15 forward and backward demonstrations for every environment to help with exploration in these sparse reward environments. We report the average success of the forward policy every $10,000$ training steps over 10 trials. More details can be found in Appendix A.3.

**Comparisons**. We compare MEDAL++ to four methods: (1) **MEDAL** [11] uses a backward policy that matches the expert state distribution by minimizing $JS(\rho^b(s) \mid\mid \rho^*(s))$, similar to ours. However, the method is designed for low-dimensional states and policy/$Q$-value networks and cannot be naïvely extended to RGB observations. For a better comparison, we improve the method to use a visual encoder with random crop and shift augmentations during training, similar to MEDAL++. (2) **R3L** [4] uses a perturbation controller as backward policy which optimizes for state-novelty computed using random network distillation [55]. Unlike our method, R3L also requires a separately collected dataset of environment observations to pre-train a VAE [56] based visual encoder, which is frozen throughout the training thereafter. (3) We consider an **oracle RL** method that trains just a forward policy and gets a privileged training environment that resets every 200 steps (i.e., the same episode length as during evaluation) and finally, (4) we consider a control method **naïve RL**, that similar to oracle trains just a forward policy, but resets every 25,000 steps similar to the non-episodic methods. We additionally report the performance of a **behavior cloning** policy, trained on the forward demonstrations used in the EARL environments. The implementation details and hyperparameters can be found in Appendix A.2.

**Results**. Figure 2 plots the evaluation performance of the forward policy versus the training samples collected in the environment. MEDAL++ outperforms all other methods on both the *sawyer* environments, and is comparable to MEDAL on *tabletop organization*, the best performing method. While R3L does recover a non-trivial performance eventually on *door closing* and *tabletop organization*, the novelty-seeking perturbation controller can cause the robot to drift to states farther away from the goal in larger environments, leading to slower improvement in evaluation performance on states starting from $s_0 \sim \rho_0$. While MEDAL and MEDAL++ have the same objective for the backward policy, optimization related improvements enable MEDAL++ to learn faster. Note, BC performs worse on *tabletop organization* environment with a 45% success rate, compared to the *sawyer* environments with a 70% and 80% success rate on *peg insertion* and *door closing* respectively. So, while BC regularization helps speed up efficiency and can lead to better policies, it can hurt the final performance of MEDAL++ if the BC policy itself has a worse success rate (at least, when true rewards are available for training, see ablations in Section 5.3). While we use the same hyperparameters for all environments, reducing the weight on BC regularization when BC policies have poor success can reduce the bias in policy learning and improve the final performance.

## 5.2 Real Robot Evaluations

In line with the main goal of this paper, our experiments aim to evaluate whether self-improvement through MEDAL++ can enable real-robots to learn more competent policies autonomously. On four manipulation tasks, we provide quantitative and qualitative comparison of the policy learned by behavior cloning on the expert data to the one learned after self-improvement by MEDAL++. We recommend viewing the results on our anonymized website in the supplementary material for training and evaluation videos, which provides a more comprehensive overview.

**Robot Setup and Tasks**. We use Franka Emika Panda arm with a Robotiq 2F-85 gripper for all our experiments. We use a RGB camera mounted on the wrist and a third person-camera, as shown in Figure 3. The final observation space includes two $100 \times 100$ RGB images, 3-dimensional end-effector position, orientation along the $z$-axis, and the width of the gripper. The action space is set up as either a 4 DoF end-effector control, or 5 DoF end-effector control with orientation along the $z$-axis depending on the task (including one degree of freedom for the gripper). Our evaluation suite consists of four manipuation tasks: grasping a cube, hanging a cloth on a hook, covering a bowl with a piece of cloth, and a (soft) peg insertion. Real world data and training is more pertinent for soft-body manipulation as they are harder to simulate, and thus, we emphasize those tasks in our evaluation suite. The tasks are shown in Figure 3.

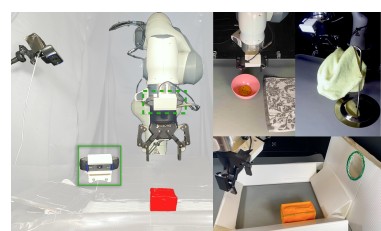

Figure 3: The training setup for MEDAL++. The image observations include a fixed third person view and a first person view from a wrist camera mounted above the gripper. The evaluation tasks going *clockwise*: cube grasping, covering a bowl with a cloth, hanging a cloth on the hook and, peg insertion.

**Training and Evaluation**. For every task, we first collect a set of 50 forward demonstrations and 50 backward demonstrations using a Xbox controller. We chain the forward and backward demonstrations to speed up collection and better approximate autonomous training thereafter. After collecting the demonstrations, the robot is trained for 30 hours using MEDAL++, collecting about $300,000$ environment transitions in the process. For the first 30 minutes of training, we reset the environment to create enough (object) diversity in the initial data collected in the replay buffer. After the initial collection, the environment is reset intermittently approximately every hour of real world training on an average, though, it is left unattended for several hours. More details related to hyperparameters, network architecture and training can be found in Appendix A.2. For evaluation, we roll-out the policy from varying initial states, and measure the success rate over 50 evaluations. To isolate the role of self-improvement, we compare the performance to a behavior cloning policy trained on the forward demonstrations using the same network architecture for the policy as MEDAL++. For both MEDAL++ and BC, we evaluate multiple intermediate checkpoints and report the success rate of the best performing checkpoint.

**Results**. The success rate of the best performing BC policy and MEDAL++ policy is reported in Table 4. MEDAL++ substantially increases the success rate of the learned policies, with approximately 30-70% improvements. We provide an abridged version of the analysis here, and defer a more detailed analysis to Appendix A.4: First, we consider a `cube-grasping` experiment. To isolate a potential source of improvement from autonomous reinforcement learning, forward demonstrations are collected from a narrow initial state distribution but the robot is evaluated starting from both in-distribution (*ID*) states and

| Task | | Behavioral Cloning | MEDAL++ |
|---|---|---|---|
| Cube Grasping | ID | 0.85 | **1.00** |
| | OOD | 0.08 | **0.82** |
| Cloth Hanging | | 0.26 | **0.62** |
| Bowl Cloth Cover | | 0.12 | **0.46** |
| Peg Insertion | | 0.04 | **0.52** |

Figure 4: Evaluation performance of the best checkpoint learned by behavior cloning and MEDAL++. Table shows the final success rates over 50 trials from randomized initial states, normalized to $[0, 1]$. MEDAL++ substantially improves the final performance compared to behavior cloning, validating the self-improvement.

out-of-distribution (*OOD*) states, visualized in Appendix, Figure 7 (*only for this experiment*). MEDAL++ improves the ID performance by 15% over the BC policy, but we see a large im-

provement of 74% on OOD performance. Autonomous training allows the robot to practice the task from a diverse set of states, including states that were OOD relative to the demonstration data. This suggests that we expect improvement in success rate to result partly from being robust to the initial state distribution, as a small set of demonstrations is unlikely to cover all possible initial states a robot can be evaluated from. Next, we evaluate MEDAL++ on grasping a cloth and putting it through a fixed hook. Here, MEDAL++ improves the success rate over BC by 36%, improving the grasp success, reducing drift and collision with the hook and importantly, reducing memorization as MEDAL++ learns a policy that re-tries grasping the cloth if it fails the first time, rather than following a memorized trajectory observed in the forward demonstrations. We observe similar trends on `bowl-covering-with-cloth` and the `peg-insertion` (5DoF) where we observe that MEDAL++ improves 34% and 48% in success rate over BC, with similar sources of improvement as `cloth-on-the-hook`. Overall, we observe that not only is MEDAL++ feasible to run in the real world with minimal task engineering, but it can also substantially improve the policy over BC.

### 5.3 Ablations

We benchmark four variants on the *tabletop organization* and *peg insertion* tasks in Figure 5: (1) MEDAL++, (2) MEDAL++ with the true reward function instead of the learned VICE reward, (3) MEDAL++ without the ensemble of $Q$-value functions, but, using SAC [57], and (4) MEDAL++ with neither BC-regularization nor oversampling expert transitions when training $Q$-value functions. The learned reward in MEDAL++ can recover or exceed the performance with true rewards. Both ensemble of $Q$-values and BC-regularization + oversampled expert transitions improve the performance, though the latter makes a larger contribution to the improvement in performance. Note, when using the true rewards, BC-regularization/oversampling expert transitions can hurt the final performance (as discussed in Section 5.1). However, when using learned rewards, they both become more important for better performance. We hypothesize that as the learned reward function becomes noisier, other components become more important for efficient learning and better final performance.

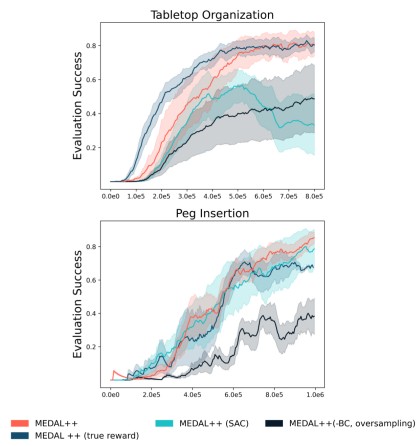

Figure 5: Ablation identifying contributions from different components of MEDAL++. Improvements from BC regularization and oversampled expert transitions are important for learning efficiency.

## 6 Discussion

We proposed MEDAL++, a method for learning autonomously from high-dimensional image observation without engineered reward functions or human oversight for repeated resetting of the environment. MEDAL++ takes a small set of forward and backward demonstrations as input, and autonomously practices the task to improve the learned policy, as evidenced by comparison with behavior cloning policies trained on just the demonstrations.

*Limitations and Future Work*: Real robot data collection is slow, even when autonomous. While the control frequency is 10 Hz, training data is collected at approximately 3.5 Hz because policy updates and collection steps are done sequentially. Asynchronous and parallel data collection and training can substantially increase the amount of data collected. Several algorithmic extensions can improve the learning efficiency: sharing the visual encoder and the environment transitions between forward and backward policies, using better network architectures and exploration specifically designed for learning autonomously can improve the sample efficiency. Additionally, reducing the number of demonstrations required per task while learning an effective reward function and minimizing the exploration challenge would lead to greater autonomy. Our work assumes the environments to be reversible. Extending MEDAL++ to environments with irreversible states, for example, using PAINT [40], is an exciting opportunity. Intermittent human interventions to reset the environment can still be important to learn successfully. The robotic system can get stuck in a specific state when collecting data autonomously due to poor exploration. Developing and using better methods for exploration or pretraining on more offline data can further reduce human interventions in training.

## 7 Acknowledgements

We would like to acknowledge Tony Zhao, Sasha Khazatsky and Suraj Nair for help with setting up robot tasks and control stack, Eric Mitchell, Joey Hejna, Suraj Nair for feedback on an early draft, Abhishek Gupta for valuable conceptual discussion, and members of IRIS and SAIL for listening to AS drone about this project on several occasions, personal and group meetings. This project was funded by ONR grants N00014-20-1-2675 and N00014-21-1-2685 and, Schmidt Futures.

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

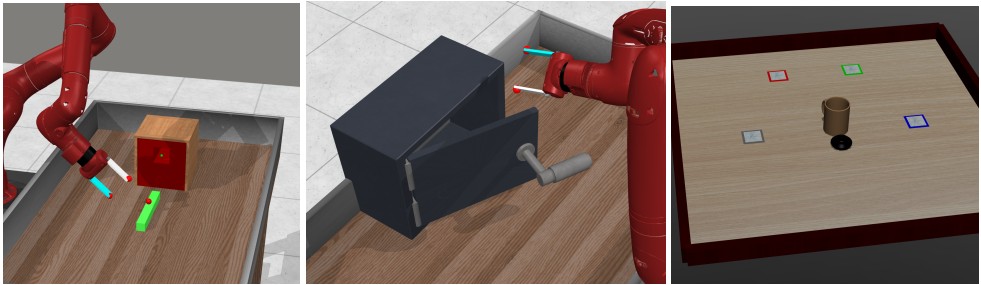

Figure 6: Environments from the EARL benchmark [5] used for simulated experiments. From left to right, the environments are: Peg insertion, Door closing and Tabletop organization.

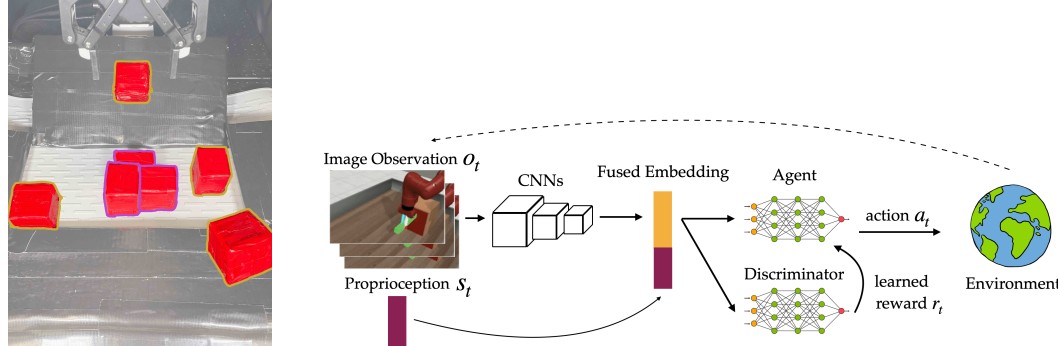

Figure 7: (*left*) Randomized position of the cube in the grasping task. The position marked by violet boundary are within the distribution of expert demonstrations, and the rest are outside the distribution. (*right*) Architecture overview for MEDAL++.

## A  Appendix

### A.1  Algorithm Overview

The pseudocode for training is given in Algorithm 1. First, the parameters and data buffers in $\mathcal{F}$ and $\mathcal{B}$ are initialized and the forward and backward demonstrations are loaded into $\mathcal{D}_f^*$ and $\mathcal{D}_b^*$ respectively. Next, we update the forward and backward goal sets, as described above. After initializing the environment, the forward policy $\pi_f$ interacts with the environment and collects data, updating the networks and buffers in $\mathcal{F}$. The control switches over to the backward policy $\pi_b$ after a fixed number of steps, and the networks and buffers in $\mathcal{B}$ are updated. The backward policy interacts for a fixed number of steps, after which the control is switched over to the forward policy and this cycle is repeated thereon. When executing in the real world, humans are allowed to intervene and reset the environment intermittently, switching the control over to $\pi_f$ after the intervention to restart the forward-backward cycle.

We now expand on how the networks are updated for $\pi_f$ during training (also visualized in Figure 9); the updates for $\pi_b$ are analogous. First, the new transition in the environment is added to $\mathcal{D}_f$. Next, we sample a batch of states from $\mathcal{D}_f$ and label them 0, and sample a batch of *equal size* from $\mathcal{D}_f^*$ and label them 1. The classifier $C_f$ is updated using gradient descent on the combined batch to minimize the cross-entropy loss. Note, the classifier is not updated for every step collected in the environment. As stated earlier, the classification problem is easier than learning the policy, and therefore, it helps to train the classifier slower than the policy. Finally, the policy $\pi_f$, $Q$-value networks $\{Q_n^f, \bar{Q}_n^f\}_{n=1}^N$ and the encoder $\mathcal{E}$ are updated on a batch of transitions constructed by sampling $(1-\rho)B$ transitions from $\mathcal{D}_f$ and $\rho B$ transitions from $\mathcal{D}_f^*$. The $Q$-value networks and the encoder are updated by minimizing $\frac{1}{N}\sum_{n=1}^N \ell(Q_n, \mathcal{E})$ (Eq 1), and the target $Q$-networks are updated as an exponential moving average of $Q$-value networks. The policy $\pi_f$ is updated by maximizing $\mathcal{L}(\pi)$. We update

**Algorithm 1:** MEDAL++

**initialize** $\mathcal{F}, \mathcal{B}$; // forward, backward parameters
$\mathcal{F}.\mathcal{D}_f^*, \mathcal{B}.\mathcal{D}_b^* \leftarrow$ `load_demonstrations()`
$\mathcal{F}.\mathcal{G}_f \leftarrow$ `get_states(`$\mathcal{F}.\mathcal{D}_f^*, -K:$`)` // last $K$ states
// exclude last $K$ states from $\mathcal{D}_f^*$, use only the last $K$ states from $\mathcal{D}_b^*$
$\mathcal{B}.\mathcal{G}_b \leftarrow$ `get_states(`$\mathcal{F}.\mathcal{D}_f^*, :-K$`)` $\cup$ `get_states(`$\mathcal{B}.\mathcal{D}_b^*, -K:$`)`
$s \sim \rho_0; \mathcal{A} \leftarrow \mathcal{F}$; // initialize environment
**while** *not done* **do**
    $a \sim \mathcal{A}$.`act(`$s$`)`$; s' \sim \mathcal{T}(\cdot \mid s, a)$;
    $\mathcal{A}$.`update_buffer(`$\{s, a, s'\}$`)`;
    $\mathcal{A}$.`update_classifier()`;
    $\mathcal{A}$.`update_parameters()`;
    // switch policy after a fixed interval
    **if** *switch* **then**
        `switch(`$\mathcal{A}, (\mathcal{F}, \mathcal{B})$`)`;
    // allow intermittent human interventions
    **if** *interrupt* **then**
        $s \sim \rho_0$;
        $\mathcal{A} \leftarrow \mathcal{F}$;
    **else**
        $s \leftarrow s'$;

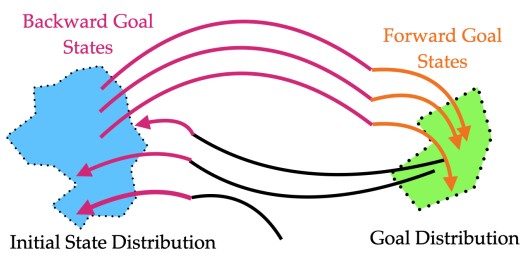

Figure 8: Visualizing the positive target states for forward classifier $C_f$ and backward classifier $C_b$ from the expert demonstrations. For forward demonstrations, last $K$ states are used for $C_f$ (*orange*) and the rest are used for $C_b$ (*pink*). For backward demonstrations, last $K$ states are used for $C_b$.

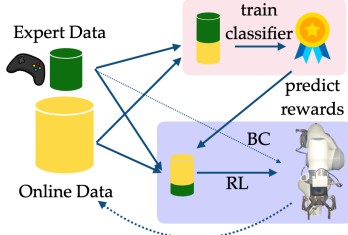

Figure 9: An overview of MEDAL++ training. The classifier is trained to discriminate states visited by an expert from the states visited online. The robot reinforcement learns on a combination of self-collected and expert transitions, and the policy learning is regularized using the behavior cloning loss.

the $Q$-value networks multiple times for every step collected in the environment, whereas the policy network is updated once for every step collected in the environment [15].

## A.2 Implementation Details and Practical Tips

An overview of the architecture used by the forward and backward networks is shown in Figure 7.

*Visual Encoder*: For the encoder, we use the same architecture as DrQ-v2 [12]: 4 convolutional layers with 32 filters of size (3, 3), stride 1, followed by ReLU non-linearities. The high-dimensional output from the CNN is embedded into a 50 dimensional feature using a fully-connected layer, followed by LayerNorm and tanh non-linearity (to output the features normalized to $[-1, 1]$). For real-robot experiments, the first person and third person views are concatenated channel wise before being passed into the encoder. The output of the encoder is fused with proprioceptive information, in this case, the end-effector position, before being passed to actor and critic networks.

*Actor and Critic Networks*: Both actor and critic networks are parameterized as 4 layer fully-connected networks with 1024 ReLU hidden units for every layer. The actor parameterizes a Gaus-

sian distribution over the actions, where a tanh non-linearity on the output restricts the actions to $[-1, 1]$. We use an ensemble size of 10 critics.

*Discriminators*: The discriminator for the forward and backward policies use a similar visual encoder but with 2 layers instead of 4. The visual embedding is passed to a fully connected network with 2 hidden layers with 256 ReLU units. When training the network, we use mixup and spectral norm regularization [53, 52] for the entire network.

*Training Hyperparameters*: For all our experiments, $K = 20$, i.e. the number of frames used as goal frames. The forward policy interacts with the environment for 200 steps, then the backward policy interacts for 200 steps. In real world experiments, we also reset the arm every 1000 steps to avoid hitting singular positions. Note, this reset does not require any human intervention as the controller just resets the arm to a fixed joint position. We use a batch size of 256 to train the policy and critic networks, out of which 64 transitions are sampled from the demonstrations (*oversampling*). We use a batch size of 512 to train the discriminators, 256 of the states come from expert data and the other 256 comes from the online data. Further, the discriminators are updated every 1000 steps collected in the environment. The update-to-data ratio, that is the number of gradient updates per transition collected in the environment is 3 for simulated environments and 1 for the real-robot experiments. We use a linearly decaying schedule for behavior cloning regularization from 1 to 0.1 over the first 50000 steps which remains fixed at 0.1 onwards throughout training.

For real world experiments, we use a wrist camera to improve the overall performance [58], and provide **only** the wrist-camera view to both discriminators. We find that this further regularizes the discriminator. Finally, we provide no proprioceptive information for the VICE discriminator, but we give MEDAL discriminator the proprioceptive information, as it needs a stronger notion of the robot's localization to adequately reset to a varied number of initial positions for improved robustness.

*Teleoperation*: To collect our demonstrations on the real robot, we use an Xbox controller that manipulates the end-effector position, orientation and the gripper state. Two salient notes: (1) The forward and backward demonstrations are collected together, one after the other and (2) the initial position for demonstrations is randomized to cover as large a state-space as feasible. The increased coverage helps with exploration during autonomous training.

## A.3 EARL Environments, Training and Evaluation

**Environments**. We consider three sparse-reward continuous-control environments from EARL benchmark [5], shown in Appendix, Fig 6). *Tabletop organization* is a simplified manipulation environment where a gripper is tasked to move the mug to one of the four coasters from a wide set of initial states, *sawyer door closing* task requires a sawyer robot arm to learn how to close a door starting from various positions, and finally the *sawyer peg insertion* task requires the sawyer robot arm to grasp the peg and insert it into a goal. Not only does the robot have to learn how to do the task (i.e. close the door or insert the peg), but it has to learn how to undo the task (i.e. open the door or remove the peg) to try task repeatedly in the non-episodic training environment. The sparse reward function is given by $r(s, a) = \mathbb{1}(\|s - g\| \leq \epsilon)$, where $g$ denotes the goal, and $\epsilon$ is the tolerance for the task to be considered completed.

**Training and Evaluation**. The environments are setup to return $84 \times 84$ RGB images as observations with a 3-dimensional action space for the *tabletop organization* (2D end-effector deltas in the XY plane and 1D for gripper) and a 4-dimensional action space for *sawyer* environments (3D end-effector delta control + 1D gripper). The training environment is reset to $s_0 \sim \rho_0$ every 25,000 steps of interaction with the environment. This is extremely infrequent compared to episodic settings where the environment is reset to the initial state distribution every 200-1000 steps. EARL comes with 5-15 forward and backward demonstrations for every environment to help with exploration in these sparse reward environments. We evaluate the forward policy every $10,000$ training steps, where the evaluation approximates $\mathbb{E}_{s_0 \sim \rho_0} \left[ \sum_{t=0}^{\infty} \gamma^t r(s_t, a_t) \right]$ by averaging the return of the

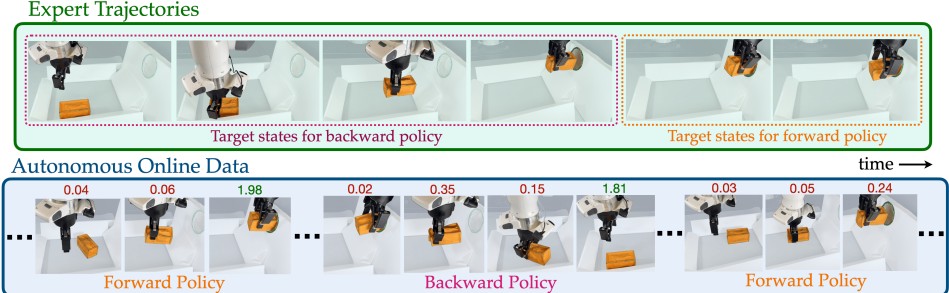

Figure 10: An overview of MEDAL++ on the task of inserting the peg into the goal location. (*top*) Starting with a set of expert trajectories, MEDAL++ learns a forward policy to *insert the peg* by matching the goal states and a backward policy to *remove and randomize the peg position* by matching the rest of the states visited by an expert. (*bottom*) Chaining the rollouts of forward and backward policies allows the robot to practice the task autonomously. The rewards indicate the similarity to their respective target states, output by a discriminator trained to classify online states from expert states.

policy over 10 episodes starting from $s_0 \sim \rho_0$. These roll-outs are used only for evaluation, and not for training.

## A.4 Real-world Experiment Analysis

We discuss the four manipulation tasks in detail. We recommend viewing the supplemental website for training and evaluation videos:

(1) *Cube Grasping*: The goal in this task is to grasp the cube from varying initial positions and configurations and raise it. For this task, we consider a controlled setting to isolate one potential source of improvement from autonomous reinforcement learning: robustness to the initial state distribution. Specifically, all the forward demonstrations are collected starting from a narrow set of initial states (*ID*), but, the robot is evaluated starting from both ID states and out-of-distribution (*OOD*) states, visualized in Appendix, Figure 7. BC policy is competent on ID states, but it performs poorly on states that are OOD. However, after autonomous self-improvement using MEDAL++, we see an improvement of 15% on ID performance, and a large improvement of 74% on OOD performance. Autonomous training allows the robot to practice the task from a diverse set of states, including states that were OOD relative to the demonstration data. This suggests that improvement in success rate results partly from being robust to the initial state distribution, as a small set of demonstrations is unlikely to cover all possible initial states a robot can be evaluated from.

(2) *Cloth on the Hook*: In this task, the robot is tasked with grasping the cloth and putting it through a fixed hook. To practice the task repeatedly, the backward policy has to remove the cloth from the hook and drop it on platform. Here, MEDAL++ improves the success rate over BC by 36%. The BC policy has several failure modes: (1) it fails to grasp the cloth, (2) it follows through with hooking because of memorization, or (3) it hits into into the hook because it drifts from the right trajectory and could not recover. Autonomous self-improvement improves upon all these issues, but particularly, it learns to re-try grasping the cloth if it fails the first time, rather than following a memorized trajectory observed in the forward demonstrations.

(3) *Bowl Covering with Cloth*: The goal of this task is to cover a bowl entirely using the cloth. The cloth can be a wide variety of initial states, ranging from 'laid out flat' to 'scrunched up' in varying locations. The task is challenging as the robot has to grasp the cloth at the correct location to successfully cover the entire bowl (partial coverage is counted as a failure). Here, MEDAL++ improves the performance over BC by 34%. The failure modes of BC are similar to previous task, including failure to grasp, memorization and failure to re-try, and incomplete coverage due to wrong initial grasp. Autonomous self-improvement substantially helps with the grasping (including retrying) and issues related to memorization. While it plans the grasps better than BC, there is room for improvement to reduce failures resulting from partially covering the bowl.

(4) *Peg Insertion*: Finally, we consider the task of inserting a peg into a goal location. The location and orientation of the peg is randomized, in service of which we use 5DoF control for this task. A successful insertion requires the toy to be perpendicular to the goal before insertion, and the error

margin for a successful insertion is small given the size of the peg and the goal. Additionally, the peg here is a soft toy, it can be grasped while being in the wrong orientation. Here, MEDAL++ improves the performance by 48% over BC. In addition to failures described in the previous tasks, a common cause of failure is the insertion itself where the agent takes an imprecise trajectory and is unable to insert the peg. After autonomous self-improvement, the robot employs an interesting strategy where it keeps retries the insertion till it succeeds. The policy is also better at grasping, though the failures of insertion often result from orienting the gripper incorrectly before the grasp which makes insertion infeasible.

