# OpenReview forum: "Self-Improving Robots: End-to-End Autonomous Visuomotor Reinforcement Learning"
_robot-learning.org/CoRL/2023/Conference — CoRL 2023 Poster_

### Official Review · Reviewer_xqXG · 2023-07-18

**Confidence:** 4
**Originality:** Fair
**Technical Quality:** Very Good
**Clarity Of Presentation:** Very Good
**Impact:** 3

**Recommendation:**

Weak Reject: I recommend rejecting the paper, but will not argue for my recommendation if the majority of other reviewers have a different opinion.

**Review:**

Strengths:

-The introduction to MEDAL is clear and concise.

-The use of VICE, among many other reward learning (e.g., inverse RL) approaches is reasonably justified.

-MEDAL++ makes MEDAL more realistic by including a convolutional autoencoder from pixel space to a state-embedding space.

-Instead of requiring a reward function, MEDAL++ learns one using VICE, which learns a classifier over goal states.

-The paper provides strong empirical results in a computational evaluation, an ablation study, and a real robot setup (compared to Behavioral Cloning).

-The paper includes a discussion of limitations.

-The paper is very well written.

-The paper includes code for reproducing the results.


Weaknesses:
-Many papers are incremental improvements over prior work. Having said that, this paper seems more of an engineering contribution than even an incremental research contribution. The swapping out of various components for other existing components brings into question whether this paper really brings a full-paper's worth of new insights. At a minimum, this paper should be a journal paper; however, it is unclear that this paper is a standalone conference paper without significantly more empirical or theoretical analysis showing the justification of these engineering choices in a way that can inform research in this area writ large.

-The proposed approach takes hundreds of thousands of episodes to outperform BC. This seems to stand in opposition to the goals of the paper of making a deployable algorithm.

-The related works section, at just two paragraphs, is a bit short considering the breadth of work in RL and LfD.

-"benchmark EARL" should have a comma before "EARL"

-"[11,4]" -- can be nice to order citations numerically

-The real-world robot evaluation does not include an RL-based baseline (e.g., MEDAL); however, this is a relatively minor point.

**Quality Of The Limitations Section:**

Limitations are addressed clearly

**Questions For Rebuttal:**

-In addition to responding to the weaknesses above, can the authors also address what kinds of problems MEDAL++ (or really VICE) would fail due to not learning a good-enough goal state classifier?

**Robotics Focus:**

Sufficient demonstration on hardware

**Summary Of Paper:**

This paper develops MEDAL++, and improvement over MEDAL, which seeks to learn robot skills through interaction with a user by learning to "do" and "undo" the task. The paper argues for the importance of removing the need for a user to reset the task to a start state. MEDAL++ improves upon MEDAL with an pixel-based encoder, a reward function-learning algorithm (VICE), and additional tweaks. The paper reports empirical results in simulation (including an ablation study) and in a real-world robot setup.

**Summary Of Recommendation:**

This paper shows positive results against a baselines that take a similar approach; however, the paper does so with incremental improvements by swapping out components of an existing baselines for existing component replacements. It is unclear that the research community would gain much additional insight into the design of these systems.

---

### Official Review · Reviewer_uNtL · 2023-07-20

**Confidence:** 4
**Originality:** Good
**Technical Quality:** Very Good
**Clarity Of Presentation:** Very Good
**Impact:** 3

**Recommendation:**

Weak Accept: I recommend accepting the paper, but will not argue for my recommendation if the majority of other reviewers have a different opinion.

**Review:**

This paper was well presented and easy to read. Several practical ideas from previous works were effectively brought together here. While there were no notable new ideas in algorithms, novelty was introduced through the online learning of complex behaviours. To train a visuo-motor policy from scratch using real-world experience and a handful of demonstrations to interact with soft objects is impressive.


**Quality Of The Limitations Section:**

Limitations are addressed clearly

**Questions For Rebuttal:**

This is a significant number of networks to update in sequence with data collection, particularly with an update-to-data ratio of 1. What is the compute used? Were there any comparative results from hyperparameter search where a smaller setup was not sufficient, e.g. ensemble number, layer number and size etc.

In Figure 5, ablation (4), is it the regularization or the oversampling that has the greater effect?

In Equation 1, should it be r + \gamma V(.)?

Similarly, the parameters and data buffers are represented by the tuple -> Similarly, the parameters and data buffers for the backward policy are represented by the tuple

can enable real-robots to be learn -> can enable real-robots to learn

regularize policy learning towards [16], -> regularize policy learning towards demonstration data [16],

as shown in Figure 4 -> should be as shown in Figure 3. Figure 4 should be Table 4 (or more preferably Table 1).

References should include their conference acceptance if applicable, e.g:
Leave no Trace: Learning to Reset for Safe and Autonomous Reinforcement Learning -> ICLR 2018

**Robotics Focus:**

Sufficient demonstration on hardware

**Summary Of Paper:**

This paper introduces MEDAL++, an extension of MEDAL. In MEDAL, forward and backwards policies perform autonomous reinforcement learning by learning to reset the environment. MEDAL++ incorporates RGB images and trains a policy end-to-end on real robot data using a small number of demonstrations. To enable this process, a host of ideas are implemented from existing works to improve efficiency, including image augmentation, ensembles of Q functions, learning reward functions from goal images, and incorporating behaviour cloning objectives.

The experiments are performed on a simulation benchmark for autonomous reinforcement learning (EARL), and on a real Franka arm performing tasks with soft objects (cloth and squishy blocks). The results show significant improvement over behaviour cloning after 30 hours of real-world interaction.


**Summary Of Recommendation:**

The implementation details of this work can be useful for others, hopefully leading towards simpler networks succeeding with fewer interactions on hardware (both demonstrations and training) and more complex tasks.

The authors efforts in the rebuttal were appreciated, sticking with the original recommendation (weak accept).

---

### Official Review · Reviewer_Y525 · 2023-07-23

**Confidence:** 4
**Originality:** Good
**Technical Quality:** Very Good
**Clarity Of Presentation:** Very Good
**Impact:** 3

**Recommendation:**

Weak Accept: I recommend accepting the paper, but will not argue for my recommendation if the majority of other reviewers have a different opinion.

**Review:**

Strength:
- Autonomous real-world robotic RL is of great important to the community, so the paper is well-motivated.
- The paper is clearly written and easy to follow.
- As a system paper, although the technical novelty is limited, the method is well evaluated on 3 real-world tasks with certain complexities that involve contacts and manipulation of deformables. It's good to see the methods working on these tasks.

Weakness:
- As mentioned, the major weakness is the lack of technical novelty, as the system is a combination & small improvements on prior work. However, this is fine when viewing the paper as a practical system.

**Quality Of The Limitations Section:**

Limitations are addressed clearly

**Questions For Rebuttal:**

The method still requires 50 demonstrations for both forward and backward policy learning. The authors are encouraged to test or add discussions on if the number of demonstrations can be further reduced.

**Robotics Focus:**

Sufficient demonstration on hardware

**Summary Of Paper:**

This paper proposes a system that can do nearly autonomous self improvement by reinforcement learning without requirement of reward engineering and frequent episodic resets. The key is to combine several techniques including MEDAL for learning a backward policy for automatic reset, VICE for learning a reward function from demonstration, drq-v2 for direct policy learning from high-dimensional pixels, and improvements on Q function learning to increase sample efficiency. The system is evaluated on 3 simulation tasks and 3 real-world tasks, and is shown to outperform several baselines.

**Summary Of Recommendation:**

As summarized above.

---

### Decision · Program_Chairs · 2023-08-30

**Decision:**

Accept (Poster)

**Comment:**

The authors propose MEDAL++ as a novel design for visuomotor RL on robotic systems. It combines a number of existing techniques to achieve this, and while technical novelty is limited, the authors show how this combination can be leveraged in a novel system design to achieve impressive real-world performance. This is also how most reviewers evaluated the paper. In addition, they appreciated the experimental evaluation on three simulated and three real-world tasks with varying complexity.